# OpenReview forum: "Improving Model Alignment Through Collective Intelligence of Open-Source Models"
_ICML.cc/2025/Conference — ICML 2025 poster_

### Official Review · Reviewer_xzNx · 2025-03-07

**Overall Recommendation:** 3

**Summary:**

This paper introduces MoAA to enhance the alignment of LLM by using collective intelligence of multiple open-source LLMs. The authors propose a 2 stage training method: the first stage uses MoA to generate diverse, high-quality synthetic SFT data, and the second stage applies DPO using MoA as a reward model. The results show significant improvements in performance on many benchmarks, including AlpacaEval2 and Arena-Hard. MoAA is shown to over models trained with data from individual LLMs, providing evidence for its effectiveness in improving alignment through synthetic data generation and preference optimization. The method also shows a self-improvement pipeline, where models fine-tuned with MoAA-generated data surpass initial capabilities.

## update after rebuttal
Checked the responses. I keep my rating.

**Claims And Evidence:**

The claims made in the paper are supported by clear and convincing evidence, because MoAA improves model alignment through synthetic data generation and preference optimization. The paper provides results showing significant performance improvements on many benchmarks. The ablation studies also show superiority of MoAA over simpler data generation methods and reward models.

**Essential References Not Discussed:**

The paper provides a strong context by citing relevant works on model alignment, synthetic data generation, and preference optimization, but it could benefit from a discussion of some recent advancements in the field of multi-agent systems and collaborative learning that are directly related to the Mixture of Agents (MoA) framework. For example, while the paper references work on Mixture of Experts (MoE) and other multi-agent frameworks, it does not cite more recent studies on dynamic multi-agent coordination and cooperative learning algorithms, such as those found in "Large Language Model Based Multi-Agents: A Survey of Progress and Challenges" (Guo et al., 2024) and "LLM-Blender: Ensembling Large Language Models" (Jiang et al., 2023). These works explore sophisticated methods for coordinating multiple models to achieve better performance, which could provide additional context for the MoAA approach presented in this paper.

**Experimental Designs Or Analyses:**

Yes, experiment design and analysis presented in paper are sound and valid.The authors use many datasets s to evaluate model performance, and 2 two-stage MoAA method is tested through a series of comparisons with other data generation and reward models. The paper also includes detailed experimental setups, hyperparameter choices, and clear performance metrics.

**Methods And Evaluation Criteria:**

Yes, the methods and evaluation criteria make sense for the problem, as they are well-aligned with the goal of improving model alignment for LLMs. The 2 stage approach, involving MoA for synthetic data generation and MoA as a reward model for preference optimization, is a logical strategy to enhance alignment by leveraging diverse open-source LLMS. The benchmark datasets  are appropriate for evaluating model performance, and their use in measuring alignment and safety ensures a comprehensive evaluation of the methods across various domains.

**Other Comments Or Suggestions:**

None

**Other Strengths And Weaknesses:**

One of the key strengths of this paper is its originality in combining existing ideas from multi-agent systems, synthetic data generation, and preference optimization to propose a scalable and effective approach for improving the alignment of large language models (LLMs). The Mixture of Agents Alignment (MoAA) method is particularly innovative in leveraging the collective intelligence of multiple open-source LLMs to generate diverse, high-quality synthetic data for model training. This approach addresses the challenge of model alignment without relying on costly proprietary models, making it both significant and practical for advancing open-source AI development. The paper is also clear in its methodology and experimental design, presenting strong empirical results that support the proposed approach. However, a potential weakness is the limited discussion of the long-term scalability of the self-improvement pipeline, which could benefit from further exploration. Overall, the paper provides valuable insights and presents a promising direction for improving LLM alignment through open-source models.
Strengths:
1. Originality: MoAA introduces a novel method to model alignment by using collective intelligence of open-source LLMs.
2. Significance: The paper addresses key challenges in model alignment, particularly the high cost and scalability issues of human-labeled data, making a strong case for the use of synthetic data generated through MoAA.
3. Clear Method: The 2 stage training process (MoAA-SFT and MoAA-DPO) is clearly outlined, providing a structured method to model alignment that is easy to follow and replicate.
Weaknesses:
1. No comparison with other approaches: Although paper compares MoAA with baseline models, it could benefit from a deeper exploration of how MoAA compares to more traditional or recently developed alignments beyond synthetic data generation.
2.Multi-turn data: The method to multi-turn instruction generation is mentioned briefly, but it lacks a detailed solution to the issue of discontinuity in multi-turn data.

**Questions For Authors:**

1. How do you ensure the diversity of the synthetic data generated by MoAA? Could biases from the participating models affect the alignment performance?
2. What are the potential computational costs associated with using the MoAA framework, especially in terms of runtime and resources required for fine-tuning?

**Relation To Broader Scientific Literature:**

The key contributions of this paper are based on and extend prior work in model alignment for LLM, particularly in synthetic data generation, multi-agent model collaboration, and preference optimization. The concept of using LLMs for data generation is from earlier work on model ensembles and MoE, is further refined in the MoAA. The paper's method uses synthetic data for SFT and DPO. However, the novelty lies in the integration of open-source LLMs into a collaborative framework, so it’s relying on models like GPT-4, and demonstrating significant improvements across benchmarks, aligning with recent trends toward using open-source LLMs for scalable AI development.

**Theoretical Claims:**

The paper doesn’t present formal proofs for theoretical claims, but instead relies on expeimental results to support its method. The claims about effectiveness of MoAA are backed by experimental evaluation, including benchmark results and ablation studies. However, the evidence provided appears robust and supports claims made in the paper without notable issues.

---

> ### Author Rebuttal · Authors · 2025-04-01
>
> We sincerely thank the reviewer for their detailed and thoughtful assessment of our work. We appreciate your recognition of the originality, significance, and methodological clarity of our MoAA framework.
>
> > The paper provides a strong context by … (Jiang et al., 2023).
> >
>
> Thank you for suggesting these valuable references. We actually cited both papers in our work.
>
> > No comparison with other … data generation.
> >
>
> Thank you for this constructive feedback. We agree that comparing with a broader range of alignment methods provides a more comprehensive evaluation of our approach. We have conducted extensive comparisons with several contemporary alignment pipelines:
>
> 1. We benchmarked against MagPie [1], which follows a similar SFT and DPO process with its generated data, and Meta-Rewarding LLM [2], an iterative alignment method utilizing self-judgment for improvement. Our method demonstrates competitive or superior performance against both approaches, as detailed in Appendix E and Table 13 of the manuscript.
> 2. As shown in Figure 4, we compared the quality of data generated by single models versus our MoA approach for MoAA-SFT. Notably, even when compared to GPT-4o-05-13, one of the most powerful closed-source models available at the time of writing, our synthesized data produced superior performance metrics on downstream tasks.
> 3. We also conducted comprehensive comparisons of different reward models within the alignment pipeline. As demonstrated in Table 6, our MoA-based judge was evaluated against individual LLM judges and specialized trained reward models. The results show that our method delivers highly competitive performance.
>
> We hope this answer your concern and are happy to answer any remaining questions!
>
> [1] [MAGPIE](https://arxiv.org/abs/2406.08464v1)
>
> [2] [Meta-Rewarding Language Models](https://arxiv.org/pdf/2407.19594)
>
> > Multi-turn data: … discontinuity in multi-turn data.
> >
>
> We thank the reviewer for rasing this concern. We design our pipeline to be able to handle the multi-turn data coherently. As briefly outlined in Section 3.1.2, our approach ensures conversational continuity through a context-aware generation process. Specifically, when generating multi-turn conversational data, we maintain coherence by providing both proposer and aggregator models with the complete conversation history. For each new turn, the previous question-answer pairs from the aggregator are included as context, allowing models to generate responses that maintain thematic consistency and proper reference resolution across turns. The aggregator has access to the full conversation history when refining proposer outputs. We have found this approach significantly reduces discontinuities in multi-turn data generation. We will include a more detailed description of this process in the revised draft.
>
> > How do you ensure the diversity … alignment performance?
> >
>
> This is a great question. Ensuring diversity in our synthetic data is essential for effective alignment. MoA inherently generates diverse outputs by combining responses from multiple LLMs with different architectures, training methods, and capabilities. This diversity in the underlying models naturally leads to varied perspectives in the generated data.
>
> As demonstrated in the original MoA paper and our Section 4.2, our approach doesn't simply combine model responses - it organically integrates and refines them. Potential biases from individual models are often corrected during aggregation, especially when using our most capable model as the aggregator.
>
> Our experimental results in Section 4.3 confirm this approach preserves diversity while enhancing data quality, with MoAA-trained models showing better generalization across diverse tasks compared to those trained on individual model data. We acknowledge that more systematic study of bias mitigation in collaborative alignment frameworks remains a valuable direction for future research.
>
> > "What are the potential computational costs … for fine-tuning?"
> >
>
> Thank you for this question! While MoA data generation requires more computation than single models, our approach remains cost-effective because:
>
> - Data generation is a one-time cost, while inference efficiency of the resulting fine-tuned model is equivalent to any similarly-sized model. As shown in Table 11, our distilled model achieving 90.6% of MoA performance at only 5.4% of the computational cost during inference.
> - Our parallel implementation runs proposer models simultaneously, significantly reducing wall-clock time.
> - Using efficient open-source models generates high-quality data without expensive proprietary APIs, with demonstrated cost savings of 23% compared to GPT-4o.
>
> Thank you again for your constructive feedback. We believe these revisions have strengthened our manuscript and addressed your concerns while further highlighting the contributions of our work.

---

### Official Review · Reviewer_nnpa · 2025-03-13

**Overall Recommendation:** 3

**Summary:**

The paper proposes Mixture of Agents Alignment (MoAA) that uses multiple LLMs to (1) generate high-quality responses for SFT training (2) provide high-quality rewards for DPO training. Experiment results show that MoAA performs better than using a single teacher LLM and the benefits are not just due to having multiple agents but also several design decisions made in MoAA. MoAA also shows potential for self-improvement, which can push the frontier of open-source LLMs without reliance on stronger external supervision.

## update after rebuttal

I have read the author response, I'll keep my score.

**Claims And Evidence:**

Yes, the claims are supported.

**Essential References Not Discussed:**

I think it's necessary to include some papers that utilize a mixture of agents for LLM as a judge such as [1].


[1] Replacing Judges with Juries: Evaluating LLM Generations with a Panel of Diverse Models

**Experimental Designs Or Analyses:**

For MoA as a reward model, I think it's important to compare to some other baselines such as those in [1].


[1] Replacing Judges with Juries: Evaluating LLM Generations with a Panel of Diverse Models

**Methods And Evaluation Criteria:**

The proposed methods and evaluation criteria make sense for the problem.

**Other Comments Or Suggestions:**

Please see other parts.

**Other Strengths And Weaknesses:**

Previous work [1] has already shown that mixture of agents could perform well in instruction following, so it's not surprising to see that their outputs could be used for (self) distillation and train a strong student model. I feel that the paper is missing some interesting explorations including how the capabilities of different models affect the final performance. If one model is significantly stronger than other models, will the method still work?


[1] Mixture-of-Agents Enhances Large Language Model Capabilities

**Questions For Authors:**

The whole MoA procesure is heavily sequential which could cost huge inference time if each model takes long to generate its output. Given that current SOTA models are mostly long CoT models, this may be an issue. Do you see any potential ways for the whole MoA process to be more time efficient?

**Relation To Broader Scientific Literature:**

Prior work has shown that a mixture of agents could perform better than single individual model in both instruction following [1] and evaluation [2]. This paper is a straightforward extension that utilizes those findings and apply them to LLM alignment.


[1] Mixture-of-Agents Enhances Large Language Model Capabilities

[2] Replacing Judges with Juries: Evaluating LLM Generations with a Panel of Diverse Models

**Theoretical Claims:**

The paper does not make theoretical claims.

---

> ### Author Rebuttal · Authors · 2025-04-01
>
> We sincerely thank the reviewer for their constructive feedback and thoughtful evaluation of our work on Mixture of Agents Alignment (MoAA). We appreciate your recognition of our paper's contributions and the validity of our claims and experimental design. We will address each concern below.
>
> > For MoA as a reward model, I think it's important to compare to some other baselines ...
> >
>
> Thank you for bringing this up! We will make sure to include and discuss the Replacing Judges with Juries paper [1] in the updated manuscript. To compare our MoA as reward model to the juries [1], we implemented the their method and evaluated it on the PPE benchmark [2]. PPE consists of 18k diverse data points spanning human preference and reasoning tasks. It achieves an average of 0.607 while our method achieves 0.661. Note that in order to make the comparison fair, we use the same models in the MoA for the panel of juries. The detailed results of the juries method and other methods are displayed below:
>
> | Model | MMLU Pro | MATH | GPQA | MBPP Plus | IFEVAL | Human Pref. | AVG |
> | --- | --- | --- | --- | --- | --- | --- | --- |
> | Gemma-2-27b-it | 0.68 | 0.73 | 0.54 | 0.58 | 0.52 | 0.6169 | 0.611 |
> | GPT-4o-mini (2024-07-18) | 0.71 | 0.81 | 0.57 | 0.54 | 0.56 | 0.6646 | 0.642 |
> | Claude-3.5 Sonnet (2024-06-20) | 0.81 | 0.86 | 0.63 | 0.54 | 0.58 | 0.6733 | 0.682 |
> | MoA as reward model | 0.76 | 0.79 | 0.58 | 0.62 | 0.57 | 0.6465 | 0.661 |
> | Juries | 0.66 | 0.67 | 0.56 | 0.61 | 0.57 | 0.57 | 0.607 |
>
> [1] [Replacing Judges with Juries](https://arxiv.org/pdf/2404.18796)
>
> [2] [How to Evaluate Reward Models](https://arxiv.org/pdf/2410.14872)
>
> > Previous work [1] has already shown that mixture of agents could  … If one model is significantly stronger than other models, will the method still work?
> >
>
> This is an excellent question. We have conducted several experiments to address this specific concern in the paper and list them here for clarity.
>
> First we include an imbalanced ensemble with one model (Gemma-2-9B-it) significantly outperforming others. Specifically, we evaluated a small-scale MoA setup (due to limited compute) with Gemma-2-9B-it, Llama-3.1-8B-Instruct, and Mistral-7B-Instruct-v0.3 as proposers, and used a two-layer MoA with Gemma-2-9B-it as the aggregator. Table 15 demonstrates that the fine-tuned gemma model shows better performance than the strongest individual model in the mix by a large margin. This finding challenges the conventional thinking that alignment requires supervision from models more capable than the target model.
>
> Second, we conducted extensive ablations examining MoA performance with varying combinations of proposers and aggregators, as documented in Table 8 and Table 15. Our analysis revealed that MoA performance serves as a reliable predictor of the distilled model's final performance, with several key insights transferring directly to MoA distillation. Notably, we found that while the ordering of proposers has minimal impact on outcomes, the capability of the aggregator model significantly influences results.
>
> Another contribution of our work is demonstrating that high-quality alignment can be achieved using exclusively open-source models. Our framework achieves competitive performance compared to strong proprietary models like GPT4-o-2024-05-13 for SFT tasks, both in terms of model performance (shown in Table 3) and cost-efficiency (with 23% cost reduction as detailed in Table 10). This finding challenges the prevailing assumption in the field that effective alignment necessarily requires access to powerful proprietary models as supervisors, potentially democratizing powerful alignment techniques for the broader research community.
>
> > The whole MoA procesure is heavily ... potential ways for the whole MoA process to be more time efficient?
> >
>
> Thank you for the feedback. This practical consideration touches on a key motivation behind our distillation framework. While data generation with MoA is indeed computationally intensive, our approach offers a valuable efficiency tradeoff: the one-time cost of MoA data generation yields a distilled model that is both efficient during inference and recovers most of the performance benefits.
>
> To address the computational challenges during data generation, we implemented batched implementation enables all proposer models to generate responses in parallel, significantly reducing runtime. We've also explored promising efficiency improvements, such as enabling the aggregator to begin generating while proposers are still completing their work, which showed encouraging early results. Although a comprehensive exploration of MoA efficiency optimizations falls beyond the scope of this work, our preliminary investigations suggest several promising directions for making the MoA process more time-efficient. We believe these efficiency considerations represent an important avenue for future research that could further enhance the practical applicability of our approach.

---

> > ### Comment · Reviewer_nnpa · 2025-04-03
> >
> > Thanks for your response. I will keep my score.

---

### Official Review · Reviewer_Jtup · 2025-03-14

**Overall Recommendation:** 3

**Summary:**

The paper demonstrates the usage of a preexisting technique, Mixture of Agents, as an alignment method. The core contribution is to use a combination of open-source LLMs as a replacement for larger proprietary models, while still obtaining competitive results with the larger proprietary models.

The authors first first distill from MoA in SFT and then generate preference pairs using MoA as a reward model to finally do DPO. They show the effectiveness of using MoA for alignment across AlpacaEval, ArenaHard and MT-Bench. They also conduct a number of experiments and ablations to study significant variables in their setup to provide a comprehensive analysis of the method.

**Claims And Evidence:**

The paper’s main claims are to be an effective alignment model for small LLMs. They make related but separate assertions for the usefulness of MoA for SFT and DPO. In both cases, they show that their results are competitive with significantly larger proprietary models through evaluations and comparative analyses.

**Essential References Not Discussed:**

N/A

**Experimental Designs Or Analyses:**

The experimental design is sound, and comparisons between models are attempted to be made in near-identical conditions

I particularly appreciated the analysis of the quality of the MoAA generated SFT and preference synthetic data.

**Methods And Evaluation Criteria:**

The proposed methods are very simple. If MoA would simply be treated as a black box, the approach used is typical for alignment pipelines. Given this, the benchmark datasets, evaluation tasks and evaluation metrics are sensible for measuring the performance of this method.

**Other Comments Or Suggestions:**

N/A

**Other Strengths And Weaknesses:**

N/A

**Questions For Authors:**

N/A

**Relation To Broader Scientific Literature:**

The key contributions of this paper heavily rely on a previous paper: “Mixture-of-Agents Enhances Large Language Model Capabilities”. The contributions of this paper are mostly derivative (i.e. they showcase the utility of MoA as an alignment technique). They also perform evaluations on the same that have significant value in demonstrating the utility of MoA for alignment.

**Theoretical Claims:**

N/A

---

> ### Author Rebuttal · Authors · 2025-04-01
>
> We sincerely thank the reviewer for their thorough evaluation of our work. We appreciate your recognition of the contributions of our alignment framework. We are pleased that you found our experimental design sound and our analysis of the quality of the MoAA-generated SFT and preference synthetic data useful. Regarding the concerns raised in your review we answer below.
>
> > The key contributions of this paper heavily rely on a previous paper: 'Mixture-of-Agents…
> >
>
> While our work builds upon the Mixture-of-Agents (MoA) framework, we believe our contributions extend significantly beyond simply showcasing MoA's utility for alignment. Our work introduces several novel aspects:
>
> 1. **Novel use of multi-LLM as a Judge**: we introduced MoA as a competitive reward model that requires zero additional training. As far as we know, not until recently, the researchers in the community started to look into using combined LLMs for model alignment. For example, [1] routes preference pairs from specific domains to domain-specific expert judges. In contrast, our MoA framework combines and refines judgments from multiple judges, utilizing their collective expertise to improve alignment. And this distinction makes our approach unique, since don’t use a router which can introduce additional bias during training. In addition, our method leverages the strengths of multiple models collaboratively rather than relying solely on isolated, domain-specific experts.
>
>     [1] Tengyu Xu, et al. (2024). The Perfect Blend: Redefining RLHF with Mixture of Judges. (online on 09/30/2024)
>
> 2. **Cost-Efficient Performance**: Our alignment method, purely relying on open-source models, is competitive against strong proprietary LLM such as GPT4-o-2024-05-13, for SFT, on both the model performances (as shown in Table 3 in our manuscript) and cost (saves 23% as shown in Table 10). This represents a potential departure from the standard paradigm where alignment has typically required access to proprietary models as supervisors.
> 3. **Strengthening the Strongest Model in MoA**: Furthermore, we tried finetuning the strongest model in the model mixtures of MoA, and still observed a clear performance boost with MoAA. We think this is a non-trivial finding because improving the strongest model in the mix provides evidence that our method can potentially push the frontier open-source models further without the supervision of stronger LLMs. Specifically, we evaluated a small-scale MoA setup (due to limited compute) with Gemma-2-9B-it, Llama-3.1-8B-Instruct, and Mistral-7B-Instruct-v0.3 as proposers, and used a two-layer MoA with Gemma-2-9B-it as the aggregator to generate the data mix. Table 15 demonstrates that the fine-tuned gemma model shows better performance than the strongest individual model in the mix by a large margin. This finding challenges the conventional wisdom that alignment requires supervision from models more capable than the target model

---

### Official Review · Reviewer_11GS · 2025-03-16

**Overall Recommendation:** 4

**Summary:**

This work proposes a novel alignment framework that uses multiple open-source LLMs within an MOA(mixture of agents) architecture to enhance model alignment via synthetic data generation (MoAA-SFT:) and preference optimization (MoAA-DPO).

Key experimental results are presented for aligning Llama-3.1-8B-Instruct and Gemma-2-9B-it and evaluating these on AlpacaEval 2, MT-Bench, and Arena-Hard benchmarks.

Strengths:
- Novel use of MOA for SFT and DPO, reduction in data generation costs for alignment compared to GPT-4o, and reduction in inference costs for aligned model in comparison to using MOA with almost similar performance.
- Extensive experimental evaluations and ablation studies.

**Claims And Evidence:**

Yes key claims are well supported by experimental results

**Essential References Not Discussed:**

None

**Experimental Designs Or Analyses:**

Yes

**Methods And Evaluation Criteria:**

Yes

**Other Comments Or Suggestions:**

None

**Other Strengths And Weaknesses:**

Although not a major weakness, the novelty may be a bit low as MOA has been previously proposed although its use in alignment is novel.

**Questions For Authors:**

None

**Relation To Broader Scientific Literature:**

The idea of using the collective knowledge of multilple open source LLMs via an MOA architecture for SFT and DPO is novel and is a valuable contribution.

**Theoretical Claims:**

None

---

> ### Author Rebuttal · Authors · 2025-04-01
>
> We sincerely thank the reviewer for the positive feedback and the supportive overall recommendation. We are pleased that you recognize our novel application of the MOA architecture for alignment through both synthetic data generation (MoAA-SFT) and preference optimization (MoAA-DPO). We appreciate your acknowledgment of the key contributions of our work, particularly the novel use of MOA for alignment tasks (both SFT and DPO), the significant reduction in data generation costs compared to GPT-4o and the decrease in inference costs for the aligned model while maintaining comparable performance.
>
> Furthermore, we agree that while the MOA architecture itself builds upon existing work, our paper's primary contribution lies in its novel application to the alignment problem, which you have acknowledged as valuable. Thank you!
>
> Thank you again for your review. We believe our work opens up exciting new directions for cost-effective alignment techniques using ensembles of open-source models.

---

### Decision · Program_Chairs · 2025-05-01

**Decision:**

Accept (poster)

**Comment:**

This paper introduces a framework leveraging multiple open-source language models for alignment. It has two-stage process: 1) generating synthetic training data and 2) employing collective judges for preference optimization.

In the rebuttal, the authors effectively addressed reviewers' concerns by: 1) demonstrating that MoAA outperforms  Replacing Judges with Juries method (2) highlighting their comprehensive ablation studies that examine how diverse model capabilities affect performance (3) clarifying their novel contribution beyond simply applying MoA to alignment, particularly emphasizing the self-improvement pipeline where models can improve without stronger external supervision, (4) explaining their implementation details for multi-turn data handling and computational efficiency optimizations. Despite building upon existing work, the novel integration of synthetic data generation, multi-agent collaboration, and preference optimization into a framework that addresses key challenges in model alignment represents a valuable contribution.

Overall, the AC recommends this paper for acceptance based on these factors: the significant performance improvements across benchmarks (doubling win rates on Arena-Hard and AlpacaEval2); cost efficiency (23% reduction compared to GPT-4o); the demonstration that open-source models can improve through collective intelligence without proprietary supervision; comprehensive experimental validation including thorough ablation studies; practical implementation details enabling reproducibility; and its potential to democratize AI development by making high-quality alignment techniques accessible to the broader research community.